# Dynamic Tensor Decomposition via Neural Diffusion-Reaction Processes

**Zheng Wang**[*]
Kahlert School of Computing
University of Utah
Salt Lake City, UT 84112
u1208847@utah.edu

**Shikai Fang**[*]
Kahlert School of Computing
University of Utah
Salt Lake City, UT 84112
shikai.fang@utah.edu

**Shibo Li**
Kahlert School of Computing
University of Utah
Salt Lake City, UT 84112
shibo@cs.utah.edu

**Shandian Zhe**[†]
Kahlert School of Computing
University of Utah
Salt Lake City, UT 84112
zhe@cs.utah.edu

## Abstract

Tensor decomposition is an important tool for multiway data analysis. In practice, the data is often sparse yet associated with rich temporal information. Existing methods, however, often under-use the time information and ignore the structural knowledge within the sparsely observed tensor entries. To overcome these limitations and to better capture the underlying temporal structure, we propose Dynamic EMbedIngs fOr dynamic Tensor dEcomposition (DEMOTE). We develop a neural diffusion-reaction process to estimate dynamic embeddings for the entities in each tensor mode. Specifically, based on the observed tensor entries, we build a multi-partite graph to encode the correlation between the entities. We construct a graph diffusion process to co-evolve the embedding trajectories of the correlated entities and use a neural network to construct a reaction process for each individual entity. In this way, our model can capture both the commonalities and personalities during the evolution of the embeddings for different entities. We then use a neural network to model the entry value as a nonlinear function of the embedding trajectories. For model estimation, we combine ODE solvers to develop a stochastic mini-batch learning algorithm. We propose a stratified sampling method to balance the cost of processing each mini-batch so as to improve the overall efficiency. We show the advantage of our approach in both simulation study and real-world applications. The code is available at https://github.com/wzhut/Dynamic-Tensor-Decomposition-via-Neural-Diffusion-Reaction-Processes.

## 1  Introduction

Multiway data is common in real-world applications and naturally represented by tensors. For example, online shopping and promotion activities can be expressed as a three-mode tensor *(customer, commodity, online merchant)*. Tensor decomposition is an important tool for multiway data analysis. It estimates embeddings for the entities in each tensor mode, with which to recover the observed

---

[*]Equal contribution

[†]Corresponding author.

37th Conference on Neural Information Processing Systems (NeurIPS 2023).

entry values. The embeddings can reflect the underlying structures within the entities and can be used as predictive features, such as for recommendation and ads auction.

In practice, tensor data is often very sparse. That is, the observed entries only take a tiny portion of all possible entries, say, $0.01\%$. In addition, the data often includes timestamps for the observed entry values, which imply rich, complex temporal variation patterns. Current tensor decomposition approaches often ignore the structure knowledge within the sparsely observed entries and under-use the temporal information, *e.g.*, simply binning the timestamps into crude time steps (Xiong et al., 2010; Rogers et al., 2013; Zhe et al., 2016a, 2015; Du et al., 2018). More important, standard tensor decomposition estimates a static embedding for each entity. However, as the representation of entities, these embeddings summarize the underlying properties of the entities, which can naturally evolve with time, such as customer interests, user income, product popularity, and fashion. Learning static embeddings can miss capturing these interesting, important temporal knowledge. While the most recent work (Wang et al., 2022) has proposed the first decomposition method to estimate embedding trajectories, it never considers the structural knowledge within the data.

To overcome these limitations, we propose DEMOTE, a dynamic embedding approach for dynamic tensor decomposition. We construct a nonlinear diffusion-reaction process in an Ordinary Differential Equation (ODE) framework to estimate embedding trajectories for tensor entities. The ODE framework is known to be flexible and convenient to handle irregularly sampled timestamps and sparsely observed data (Rubanova et al., 2019). In addition, since ODE models focus on learning the dynamics (*i.e.*, time derivatives) of the target function, they have promising potential for providing robust, accurate long-term predictions (via integration with the dynamics). Specifically, to leverage the structural knowledge within the data, we first build a multi-partite graph based on the observed entries. The graph encodes the correlations between entities at different modes in terms of their interactions. We then construct a graph diffusion process in the ODE to co-evolve the embedding trajectories of correlated entities. Next, we use a neural network to construct a reaction process to model the individual-specific evolution for each entity. In this way, our neural diffusion-reaction process captures both the commonalities and personalities of the entities in learning their dynamic embeddings. Given the embedding trajectories, we model the entry value as a latent function of the associated entities' trajectories. We use another neural network to flexibly estimate the function and to capture the complex relationships of the entities. For efficient training, we base on ODE solvers to develop a stochastic mini-batch learning algorithm. We develop a stratified sampling scheme, which can balance the cost of executing the ODE solvers in each mini-batch so as to improve the efficiency.

We evaluated our method in both simulation and real-world applications. The simulation experiments show that DEMOTE can successfully capture the underlying dynamics of the entities from their temporal interactions and recover the hidden clustering structures within the trajectories. Then in three real-world applications, we tested the accuracy in predicting the tensor entry values at different time points. DEMOTE consistently outperforms the state-of-the-art decomposition methods that incorporate temporal information, often by a large margin. We also demonstrated that both the diffusion and reaction processes contribute to the learning performance. Finally, we investigated the learned embedding trajectories and found interesting evolution paths and hidden structures.

## 2 Notations and Background

Suppose we have collected data for a $K$-mode tensor. Each mode $k$ includes $d_k$ entities, which we index by $1, \ldots, d_k$. We then index each tensor entry by a tuple $\boldsymbol{\ell} = (l_1, \ldots, l_K)$ where for each $k$, we have $1 \le l_k \le d_k$. Suppose we observed $N$ tensor entry values and timestamps. The dataset is denoted by $\mathcal{D} = \{(\boldsymbol{\ell}_1, t_1, y_1), \ldots, (\boldsymbol{\ell}_N, t_N, y_N)\}$ where $\{t_n\}$ and $\{y_n\}$ are the timestamps and entry values, respectively. Our goal is for each entity $j$ of mode $k$, to estimate a dynamic embedding $\mathbf{u}_j^k(t) : \mathbb{R}_+ \to \mathbb{R}^R$. That is, the embedding is a time function (trajectory) with an $R$-dimensional output. Standard tensor decomposition introduces a static embedding representation for each entity, namely, $\mathbf{u}_j^k$ is considered as time invariant. Tensor decomposition aims to estimate the embeddings (or factors) to reconstruct the tensor. For example, the classical Tucker decomposition (Tucker, 1966) employs a multilinear factorization model, $\mathcal{M} = \mathcal{W} \times_1 \mathbf{U}^1 \times_2 \ldots \times_K \mathbf{U}^K$, where $\mathcal{M} \in \mathbb{R}^{d_1 \times \ldots \times d_k}$ is the entire tensor, $\mathcal{W} \in \mathbb{R}^{R_1 \times \cdots \times R_K}$ is the tensor-core parameter, $\mathbf{U}^k$ comprises all the embeddings of the entities in mode $k$, and $\times_k$ is the tensor-matrix product at mode $k$ (Kolda, 2006). The popular CANDECOMP/PARAFAC (CP) decomposition (Harshman, 1970) can be viewed as a simplified version of Tucker decomposition, where we set $R_1 = \ldots = R_K = R$ and the tensor-core $\mathcal{W}$ to be

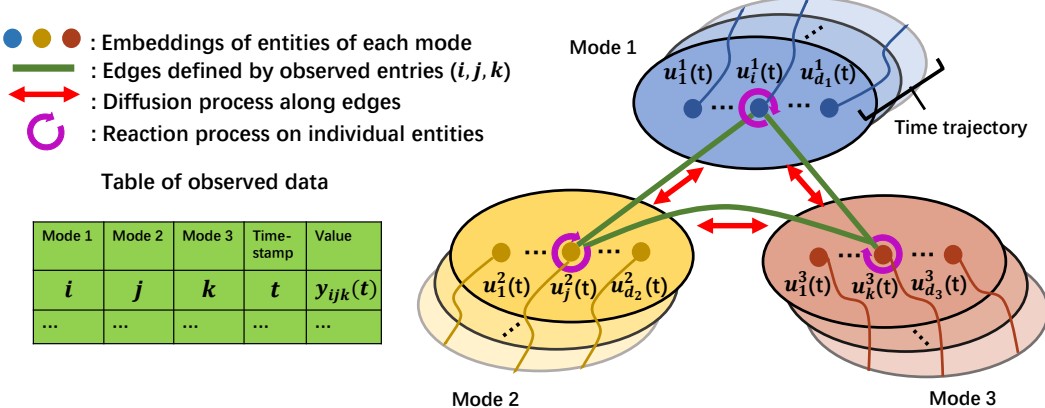

Figure 1: The illustration of the embedding model in DEMOTE.

diagonal. Hence, each entry value is factorized as $m_{\boldsymbol{\ell}} = (\mathbf{u}_{l_1}^1 \circ \ldots \circ \mathbf{u}_{l_K}^K)^\top \boldsymbol{\lambda}$, where $\circ$ is the Hadamard (element-wise) product, and $\boldsymbol{\lambda}$ corresponds to $\text{diag}(\mathcal{W})$. While CP and Tucker decomposition are popular, their multilinear modeling can be oversimplistic for complex applications. To estimate nonlinear relationships of the entities, Xu et al. (2012); Zhe et al. (2015, 2016a) used a Gaussian process (GP) (Rasmussen and Williams, 2006) to model the entry value as a random function of the embeddings, $m_{\boldsymbol{\ell}} = g(\mathbf{u}_{l_1}^1, \ldots, \mathbf{u}_{l_K}^K)$, where $g \sim \mathcal{GP}(0, \kappa(\mathbf{x}_{\boldsymbol{\ell}}, \mathbf{x}_{\boldsymbol{\ell}'}))$, $\mathbf{x}_{\boldsymbol{\ell}} = [\mathbf{u}_{l_1}^1; \ldots; \mathbf{u}_{l_K}^K]$ and $\mathbf{x}_{\boldsymbol{\ell}'} = [\mathbf{u}_{l_1'}^1; \ldots; \mathbf{u}_{l_K'}^K]$ are the embeddings of the entities in entry $\boldsymbol{\ell}$ and $\boldsymbol{\ell}'$, respectively, and $\kappa(\cdot, \cdot)$ is the covariance (kernel) function. Given the GP prior, any finite set of $N$ entry values follow a multi-variate Gaussian distribution, $\mathbf{m} \sim \mathcal{N}(\mathbf{0}, \mathbf{K})$, where $\mathbf{m} = [m_{\boldsymbol{\ell}_1}, \ldots, m_{\boldsymbol{\ell}_N}]$, $\mathbf{K}$ is the $N \times N$ kernel matrix, and each $[\mathbf{K}]_{i,j} = \kappa(\mathbf{x}_{\boldsymbol{\ell}}, \mathbf{x}_{\boldsymbol{\ell}'})$. Suppose we have collected continuous observations for the $N$ entries $\mathbf{y} = [y_1, \ldots, y_N]$. We can use a Gaussian noise model: $y_n = m_{\boldsymbol{\ell}_n} + \epsilon_n$ where $\epsilon_n \sim \mathcal{N}(0, \sigma^2)$. The marginal likelihood is $p(\mathbf{y}) = \mathcal{N}(\mathbf{y}|\mathbf{0}, \mathbf{K} + \sigma^2 \mathbf{I})$, which we can maximize to estimate the embeddings and the model parameters.

Practical data often includes temporal information, *i.e.*, the timestamp when each observed entry value is generated. To leverage this information, existing methods often bin the timestamps into a series of steps, say, by weeks or months (Xiong et al., 2010; Rogers et al., 2013; Zhe et al., 2016a; Song et al., 2017). The tensor is then expanded with an additional time-step mode, and one can apply any decomposition algorithm to estimate embeddings for both the entities and time steps. To capture the temporal dependency, a conditional model is often used (Xiong et al., 2010), say, $p(\mathbf{t}_{j+1}|\mathbf{t}_j) = \mathcal{N}(\mathbf{t}_{j+1}|\mathbf{t}_j, \tau\mathbf{I})$ where $\mathbf{t}_j$ is the embedding of $j$-th step. To leverage the continuous time information, Zhang et al. (2021) recently developed continuous CP decomposition, where the coefficients $\boldsymbol{\lambda}$ are modeled as a time function with polynomial splines.

## 3 Model

Standard tensor decomposition assumes the embeddings are static and time-invariant. However, the embeddings summarize and extract the properties of entities, which can evolve with time, such as customer interests, health status, and product popularity. Therefore, only estimating static embeddings can miss important temporal variations of the entities' properties, resulting in poor representations and predictive performance. In addition, practical tensor data are typical sparse, and only a small portion of entries actually have data. Within these entries can be valuable structural knowledge. Current methods, however, are rarely aware of such knowledge. To overcome these limitations, we propose DEMOTE, a novel dynamic embedding approach.

Specifically, we propose an ODE model to learn the embedding trajectories $\{\mathbf{u}_j^k(t) | 1 \le k \le K, 1 \le j \le d_k\}$. The ODE framework is known to be amenable for irregularly sampled, sparsely observed data. More important, ODE models concentrate on learning the time derivative $\mathrm{d}\mathbf{u}_j^k/\mathrm{d}t$ (*i.e.*, dynamics), rather than the trajectory function itself. Therefore, they have a promising potential to give reliable, long-term trajectory prediction (via numerical integration) even at time points far away from the training timestamps, provided the time derivative is well captured. We construct a joint ODE model for all the embedding trajectories. The ODE consists of a diffusion process and a

reaction process. The diffusion process leverages the structural knowledge in data to co-evolve the embeddings of correlated entities, so as to better overcome the data sparsity. The reaction process models the entity-specific evolution so that it can capture the individual differences in the embedding evolution. The ODE model synergizes the two processes to capture both the commonalities and personalities of these embedding trajectories.

**Diffusion Process on Multi-Partite Graphs.** First, we construct a graph-based diffusion process to exploit the entity correlations reflected in data. Intuitively, if an observed entry involves entity A (*e.g.*, customer A) and B (*e.g.*, commodity B), the two entities are likely correlated. Thus, we can draw an edge between A and B to express the correlation. We then generalize this intuition to create a $K$-partite undirected graph $\mathbb{G}(E, V)$, to encode such correlations across all the entities in the $K$ tensor modes. Each vertex represents a particular entity, and the entire collection of the entities is partitioned into $K$ groups, $V = V^1 \cup \ldots \cup V^K$, where group $V^k = \{v_1^k, \ldots, v_{d_k}^k\}$ represents the entities of mode $k$. Two entities (at different modes) are connected if they were observed to interact, namely, $(v_j^k, v_{j'}^{k'}) \in \mathbb{E}$ if $\exists \ell_n \in \mathcal{D}$ such that $\ell_n = (\ldots, j, \ldots, j', \ldots)$ where $j$ and $j'$ are indices at mode $k$ and $k'$, respectively. See Fig. 1 for an illustration. This graph naturally implies underlying information diffusion across the entities within their interactions. For example, if customer A connects to products B and C, it might mean that A distributes their interests/willingness/budgets to purchase B and C. The edges between one merchant A and a list of products {B, C, ... } might indicate the diffusion of willingness to increase the inventory of these products.

To flexibly estimate the diffusion rate, we introduce a weight $w_{j,j'}^{k,k'}$ for each edge $(v_j^k, v_{j'}^{k'}) \in E$. We then arrange these weights into $K(K-1)$ adjacent matrices, $\mathcal{W} = \{\mathbf{W}^{k,k'} | 1 \leq k, k' \leq K, k \neq k'\}$ where $\mathbf{W}^{k,k'} = \left(\mathbf{W}^{k',k}\right)^\top$. Each $\mathbf{W}^{k,k'}$ is a sparse $d_k \times d_{k'}$ matrix that represents the edges and edge weights between $V^k$ and $V^{k'}$, *i.e.*, $[\mathbf{W}^{k,k'}]_{j,j'} = w_{j,j'}^{k,k'}$ if $(v_j^k, v_{j'}^{k'}) \in E$ and 0 otherwise. We now construct a diffusion process based on the $K$-partite graph. We view the embedding trajectory as a kind of concentration. For each entity $j$ at mode $k$, the change rate of its concentration (embedding) $\mathbf{u}_j^k(t)$ is determined by the difference from the concentrations of its neighbors. Since the neighbors come from entities of all the other $K - 1$ modes, we have

$$\frac{\mathrm{d}\mathbf{u}_j^k}{\mathrm{d}t} = \sum_{s \in \{1,\ldots,K\} \setminus k} \sum_{j'=1}^{d_s} [\mathbf{W}^{k,s}]_{j,j'} \left(\mathbf{u}_{j'}^s(t) - \mathbf{u}_j^k(t)\right) = \sum_{s \in \{1,\ldots,K\} \setminus k} \left(\mathbf{w}_j^{k,s} \mathbf{U}^s(t)\right)^\top - a_j^{k,s} \mathbf{u}_j^k,$$

where $\mathbf{w}_j^{k,s}$ is the $j$-th row of $\mathbf{W}^{k,s}$, $\mathbf{U}^s(t) = [\mathbf{u}_1^s(t), \ldots, \mathbf{u}_{d_s}^s(t)]^\top$ is the embeddings of all the entities at mode $s$, of size $d_s \times R$, and $a_j^{k,s} = \sum_{j'=1}^{d_s} [\mathbf{W}^{k,s}]_{j,j'}$ is the degree of vertex $j$ in $\mathbf{W}^{k,s}$. We can see that the evolution of the embeddings for different modes are coupled. Hence, it is natural to formulate the diffusion process jointly for all the embeddings, $\frac{\mathrm{d}\mathcal{U}(t)}{\partial t} = \mathrm{d}\left(\mathbf{U}^1(t), \ldots, \mathbf{U}^K(t)\right) / \mathrm{d}t = \mathcal{W}\mathcal{U}(t) - \mathcal{A}\mathcal{U}(t) = (\mathcal{W} - \mathcal{A})\mathcal{U}(t)$ where

$$\mathcal{W} = \begin{pmatrix} \mathbf{0} & \mathbf{W}^{1,2} & \ldots & \mathbf{W}^{1,K} \\ \mathbf{W}^{2,1} & \mathbf{0} & \ldots & \vdots \\ \vdots & & \ddots & \mathbf{W}^{K-1,K} \\ \mathbf{W}^{K,1} & \ldots & \mathbf{W}^{K,K-1} & \mathbf{0} \end{pmatrix},$$

$\mathcal{A} = \mathrm{diag}\left(\sum_{s \in \{1\ldots K\} \setminus 1} \mathbf{A}^{1,s}, \ldots, \sum_{s \in \{1\ldots K\} \setminus K} \mathbf{A}^{K,s}\right)$, and each $\mathbf{A}^{k,s} = \mathrm{diag}(a_1^{k,s}, \ldots, a_{d_k}^{k,s})$ is the degree matrix of $\mathbf{W}^{k,s}$.

**Reaction Process of Individual Entities.** Next, to capture the individual difference of each entity in evolving their embeddings, we model a local reaction process for each entity, $\mathbf{f}_{\boldsymbol{\theta}_k}(\mathbf{u}_j^k(t), t)$, where $\mathbf{f}(\cdot)$ is a neural network (NN), and $\boldsymbol{\theta}_k$ are the NN (reaction) parameters for mode-$k$ entities. The metaphor from the chemical physics is as follows. While substances are being diffused across different sites, at each site a chemical reaction process happens concurrently, which varies the concentration locally. We extend the model as a joint diffusion-reaction process,

$$\frac{\partial \mathcal{U}(t)}{\partial t} = (\mathcal{W} - \mathcal{A})\mathcal{U}(t) + \mathcal{F}(\mathcal{U}, t), \quad \mathcal{U}(0) = \mathcal{U}_0, \tag{1}$$

where $\mathcal{F}(\mathcal{U}, t) = [\mathbf{f}_{\boldsymbol{\theta}_1}(\mathbf{u}_1^1, t), \ldots, \mathbf{f}_{\boldsymbol{\theta}_1}(\mathbf{u}_{d_1}^1, t), \ldots, \mathbf{f}_{\boldsymbol{\theta}_K}(\mathbf{u}_1^K, t), \ldots, \mathbf{f}_{\boldsymbol{\theta}_K}(\mathbf{u}_{d_K}^K, t)]^\top$.

**Entry Value Generation.** Given the embedding trajectories, to obtain the tensor entry value $m_{\boldsymbol{\ell}}$ at arbitrary time $t$, we model $m_{\boldsymbol{\ell}}(t)$ as a function of the relevant embeddings at time $t$,

$$m_{\boldsymbol{\ell}}(t) = g\left(\mathbf{u}_{l_1}^1(t), \ldots, \mathbf{u}_{l_K}^K(t)\right). \tag{2}$$

While one can follow (Xu et al., 2012; Zhe et al., 2016b) to assign a GP prior over $g(\cdot)$, the GP model needs to compute a giant kernel matrix over all the observed entry values (see Sec. 2). It is computationally too expensive or infeasible when the number of observations is large. Hence one has to seek for complex low-rank approximations. To avoid this problem, we model $g$ with another neural network, which is not only as flexible as GP, but is more scalable and convenient for computation. Since now, the input to $g(\cdot)$ consists of the trajectory values, which vary with time, our NN model for $g$ can flexibly capture the complex temporal relationship between the entities. We finally sample the observed entry values with a Gaussian noise model, $p(\mathbf{y}|\mathbf{m}) = \mathcal{N}(\mathbf{y}|\mathbf{m}, \sigma^2\mathbf{I})$ where $\mathbf{y} = [y_1, \ldots, y_N]^\top$ and $\mathbf{m} = [m_{\boldsymbol{\ell}_1}(t_1), \ldots, m_{\boldsymbol{\ell}_N}(t_N)]^\top$. We focus on real-valued data in this paper. However, it is straightforward to extend our approach to other types of data.

## 4 Model Estimation

Given data $\mathcal{D} = \{(\boldsymbol{\ell}_1, t_1, y_1), \ldots, (\boldsymbol{\ell}_N, t_N, y_N)\}$, the joint probability of our model is

$$p(\boldsymbol{\beta}, \{\boldsymbol{\theta}_k\}, \mathbf{y}) = p(\boldsymbol{\beta}) \cdot \prod_{k=1}^K p(\boldsymbol{\theta}_k) \cdot \prod_{n=1}^N \mathcal{N}\left(y_n | g\left(\mathbf{u}_{l_{n1}}^1(t_n), \ldots, \mathbf{u}_{l_{nK}}^K(t_n)\right), \sigma^2\mathbf{I}\right), \tag{3}$$

where $\boldsymbol{\beta}$ is the NN parameters for $g$, each $\boldsymbol{\theta}_k$ is the NN reaction parameters for mode-$k$ entities, $p(\boldsymbol{\beta})$ and $p(\boldsymbol{\theta}_k)$ are element-wise standard Gaussian, and $\mathbf{y} = (y_1, \ldots, y_N)^\top$. To obtain the trajectory values in the Gaussian likelihood of each $y_n$, we need to solve the ODE in (1) to time $t_n$,

$$\mathcal{U}(t_n) = \text{ODESolve}(\mathcal{U}_0, 0, t_n, \Theta) \tag{4}$$

where $\Theta = \{\mathcal{W}, \boldsymbol{\theta}_1, \ldots, \boldsymbol{\theta}_K\}$ consists of the ODE parameters. Our goal is to estimate $\Theta$, the initial state $\mathcal{U}_0$, the NN parameters $\boldsymbol{\beta}$, and the noise variance $\sigma^2$.

**Stratified Mini-Batch Sampling.** We use stochastic mini-batch optimization to maximize the log joint probability so as to estimate all the required parameters,

$$\mathcal{L} = \log p(\boldsymbol{\beta}, \{\boldsymbol{\theta}_k\}, \mathbf{y}) = \log(\text{Prior}) - \sum_{n=1}^N \log \mathcal{N}\left(y_n | g\left(\mathbf{x}_n\right), \sigma^2\mathbf{I}\right)$$

where $\log(\text{Prior}) = \log p(\boldsymbol{\beta}) + \sum_{k=1}^K \log p(\boldsymbol{\theta}_k)$, and $\mathbf{x}_n = \left(\mathbf{u}_{l_{n1}}^1(t_n), \ldots, \mathbf{u}_{l_{nK}}^K(t_n)\right)$. Each time, we sample a mini-batch of observations $\mathcal{B}$, and obtain an unbiased stochastic estimate of the log probability, $\widehat{\mathcal{L}} = \log(\text{Prior}) - \frac{N}{B} \sum_{n \in \mathcal{B}} \left[\log \mathcal{N}(y_n | g(\mathbf{x}_n), \sigma^2)\right]$. We compute $\nabla\widehat{\mathcal{L}}$ as the stochastic gradient to update all the parameters.

For each data point $n$ in the mini-batch, we need to run ODE solving (4) to obtain $\mathbf{x}_n = \left(\mathbf{u}_{l_{n1}}^1(t_n), \ldots, \mathbf{u}_{l_{nK}}^K(t_n)\right)$. To back-propagate the gradient so as to compute the gradient w.r.t the ODE parameters $\Theta$ and initial state $\mathcal{U}_0$, we can either construct a computational graph during the running of the solver (*e.g.*, the Runge-Kutta method (Dormand and Prince, 1980)), or use the adjoint state method (Pontryagin, 1987; Chen et al., 2018) that solves an adjoint backward ODE to compute the gradient. In whichever case, we need to sort the time points in the mini-batch and solve the ODE sequentially for these time points. As a result, the number of *unique* time points in the mini-batch greatly influences the speed of processing the mini-batch. The standard mini-batch sampling (based on the training example indices) can result in an uneven allocation of the computational cost across the mini-batches — some mini-batch is fast and some including more unique time points is much slower. To address this issue, we use a simple stratified sampling approach.

- We collect the unique time points in the whole dataset, $\mathcal{T} = \{\tau_1, \tau_2, \ldots\}$ at the beginning.
- To conduct each stochastic update, we first sample $B$ unique time points $\mathcal{C}$ from $\mathcal{T}$, then for each time point $\tau_j \in \mathcal{C}$, we look at all the observed entry values produced at $\tau_j$, namely $\mathcal{D}_{\tau_j} = \{(\boldsymbol{\ell}_n, t_n, y_n) \in \mathcal{D} | t_n = \tau_j\}$.
- We randomly sample one example from each $\mathcal{D}_{\tau_j}$ to collect the mini-batch $\mathcal{B}$.

In this way, we ensure the cost of running ODE solvers and related gradient computation in each mini-batch is identical. There are no fluctuations in cost/running time when processing different min-batches. Empirically, we found that the overall speed of our method is much faster than vanilla stochastic mini-batch optimization (see Table 2 in Appendix).

## 5    Related Work

There have been many works on tensor decomposition, such as (Yang and Dunson, 2013; Rai et al., 2014; Zhe et al., 2015, 2016b,a; Tillinghast et al., 2020; Pan et al., 2020b; Fang et al., 2021a,b; Tillinghast and Zhe, 2021; Tillinghast et al., 2022; Fang et al., 2022). To integrate temporal information, most approaches augment the tensor with a time mode (Xiong et al., 2010; Rogers et al., 2013; Zhe et al., 2016b; Ahn et al., 2021; Zhe et al., 2015; Du et al., 2018), which includes a list of time steps. To estimate the temporal dependencies, existing methods often employ a dynamic model over the time steps, such as a conditional Gaussian prior (Xiong et al., 2010), recurrent neural networks(Wu et al., 2019), and kernel smoothing/regularization (Ahn et al., 2021). To leverage continuous timestamps, Zhang et al. (2021) modeled the CP coefficients as a time function, and Fang et al. (2022) modeled the tensor-core as a time function in the Tucker decomposition. The most recent work (Wang et al., 2022) places a GP prior in the frequency domain, and construct a bi-level GP model to learn factor or embedding trajectories as a combination of Fourier bases.

Another set of works factorize the interaction *events* (Schein et al., 2015, 2016; Zhe and Du, 2018; Pan et al., 2020a; Wang et al., 2020; Pan et al., 2021; Wang et al., 2022), and they *cannot* predict the interaction results (*e.g.*, tensor entry values like purchase amount and product ratings). These works mainly leverage Poisson processes, Hawkes processes, or more general point processes to estimate the event rate. Like the standard tensor factorization, these methods also estimate static embeddings for the event participants.

Our model can be viewed as an extension of the neural ODE model (Chen et al., 2018). If we only employ the reaction process for each entity, our model is a latent neural ODE (we have an additional NN that combines the latent trajectories to predict the tensor entry values). However, we further leverage the structural knowledge in data to construct a multi-partite graph so as to encode the correlations of the entities. Based on the multi-partite graph, we construct a diffusion process to co-evolve the embeddings of the entities. In doing so, we can better overcome the data sparsity issue. The most recent work (Li et al., 2022) uses a neural ODE to model the tensor *entry value* as a function of the involved embeddings and time. It differs from our method in that *(1) it still learns a static (time-invariant) embedding for each entity, and (2) its modeling does not use the structure knowledge within the tensor data like our method*. Many other works have developed graph diffusion processes based on graph data. For example, Chamberlain et al. (2021) proposed a graph neural network (GNN) by using multi-head attention to construct the adjacent matrix for a graph diffusion equation over the graph nodes. Atwood and Towsley (2016) introduced a diffusion operator to develop diffusion-convolutional neural networks. Huang et al. (2021) developed a GNN-based ODE to model both the nodes and edges in dynamic graphs.

## 6    Experiment

### 6.1    Simulation Study

We first examined DEMOTE on a synthetic task. Specifically, we considered a two-mode tensor, where each mode includes 20 entities. Each entity has one underlying embedding trajectory. In the first mode, the trajectory of each entity is an exponential function, $u_j^1(t) = c_j^1 \exp(0.5 c_j^1 t)$ ($1 \le j \le 20$), while in the second mode is a linear function, $u_j^2(t) = c_j^2 + 2\pi c_j^2 t$. We generated two clusters of trajectories for each mode, where those of the first ten entities form the first cluster and the remaining the second cluster. To this end, for mode 1, we sampled the coefficients of the first ten entities' trajectories, namely, $[c_1^1, \ldots, c_{10}^1]^\top$, from $\mathcal{N}\left([-5, \ldots, -5]^\top, 0.1\mathbf{I}\right)$, and the remaining ten coefficients $[c_{11}^1, \ldots, c_{20}^1]^\top$ from $\mathcal{N}\left([0.5, \ldots, 0.5]^\top, 0.1\mathbf{I}\right)$. Then for mode 2, we sampled each coefficient $c_j^2$ conditioned on its counter-part for mode 1, $c_j^2|c_j^1 \sim \mathcal{N}(c_j^2|c_j^1, 0.1)$. That means, the coefficients of cluster-1 trajectories across the two modes are close, and so are those of cluster-2 trajectories. The value for a particular entry $\boldsymbol{\ell} = (l_1, l_2)$ is generated by

$$m_{\boldsymbol{\ell}}(t) = \left(u_{l_1}^1(t)\right)^{\mathbb{1}(l_1+l_2 \mod 2=0)} \cdot \left(u_{l_2}^2(t)\right)^{\mathbb{1}(l_1+l_2 \mod 2=1)} \tag{5}$$

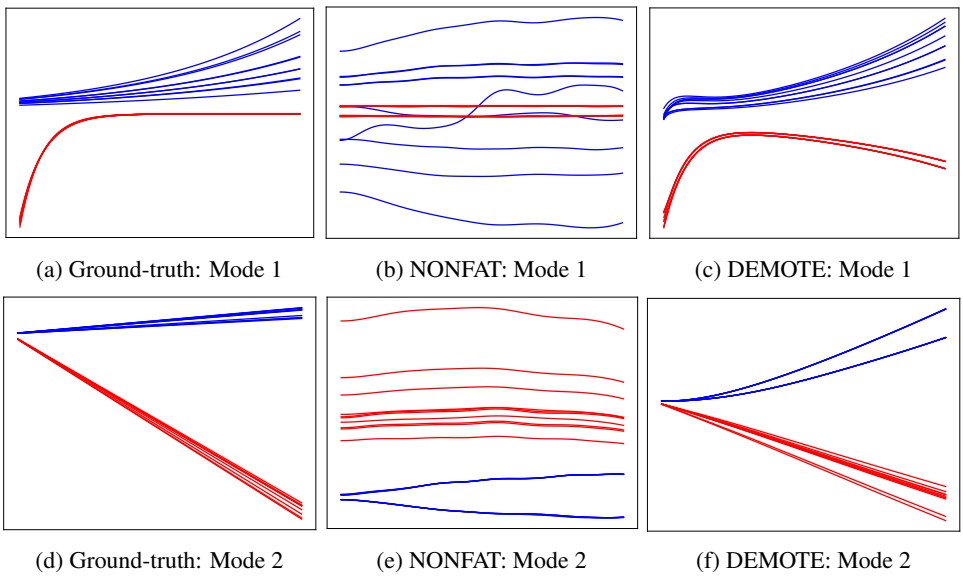

| (a) Ground-truth: Mode 1 | (b) NONFAT: Mode 1 | (c) DEMOTE: Mode 1 |
| (d) Ground-truth: Mode 2 | (e) NONFAT: Mode 2 | (f) DEMOTE: Mode 2 |

Figure 2: The estimated embedding trajectories for each mode. The color indicates the ground-truth cluster membership.

where $\mathbb{1}(\cdot)$ is the indicator function. When $l_1 + l_2$ is even, the entry value is the trajectory value of the first entity; otherwise, it is the trajectory value of the second entity. To generate the training data, we randomly sampled entries from $\{(l_1, l_2)|1 \leq l_1, l_2 \leq 10\} \cup \{(l_1, l_2)|11 \leq l_1, l_2 \leq 20\}$ (namely, interactions between cluster-1 entities of the two modes, and between cluster-2 entities). We then sampled $t \sim \text{Unifrom}[0, 5]$, to obtain the corresponding entry values. We randomly generated 6,400 entry values and the timestamps for training, and another 1,600 data points for testing.

We implemented our method with Pytorch (Paszke et al., 2019). We used `torchdiffeq` library (`https://github.com/rtqichen/torchdiffeq`) to solve ODEs and to compute the gradient w.r.t ODE parameters and initial states via automatic differentiation. For the NN of the reaction process, we used one hidden layer, with 10 neurons and tanh activation, and for the NN to predict the interaction result, we used two hidden layers, 50 neurons per layer and tanh activation.

We compared with NONFAT (NONparametric Factor Trajectory learning) (Wang et al., 2022), a bi-level latent GP model that uses Fourier bases to estimate factor trajectories for dynamic tensor decomposition. To our knowledge, this work is the only method (and also the most recent) that estimates trajectories. We used the original implementation (`https://github.com/wzhut/NONFAT`) and the default settings. We set the mini-batch size to 50, and used ADAM (Kingma and Ba, 2014) algorithm for stochastic optimization. The learning rate was automatically adjusted in $[10^{-4}, 10^{-1}]$ by the `ReduceLROnPlateau` scheduler (Al-Kababji et al., 2022). The maximum number of epochs is 2K, which is enough for convergence. The estimated trajectories are shown in Fig. 2a-f. As we can see, our estimation (Fig. 2c and 2f) well matches the ground-truth and accurately recovers the cluster structure of the trajectories. The root-mean-square error (RMSE) on the test set is 0.032. By contrast, although the test error of NONFAT is close to DEMOTE (0.034), its learned trajectories (Fig. 2b and 2e) are far from the ground-truth, and fail to reflect the cluster structure. These have shown the advantage of DEMOTE in capturing complex relationships within data to recover the underlying trajectories and their structure.

## 6.2 Prediction Accuracy

**Datasets.** We next evaluated the predictive performance of DEMOTE in three real-world applications. (1) *CA Weather* (Moosavi et al., 2019) (`https://smoosavi.org/datasets/lstw`), weather conditions in California from August 2016 to December 2020. We extracted a four-mode tensor for 7 different weather *types*, 6 *severity levels*, 30 *latitudes* and 30 *longitudes* in GPS coordinates. The entry value is the count of the particular weather condition. We collected 15K observed tensor entry values and the timestamps. (2) *CA Traffic* (Moosavi et al., 2019) (`https://smoosavi.org/datasets/lstw`), traffic accidents in California from January 2018 to December 2020. We extracted a four

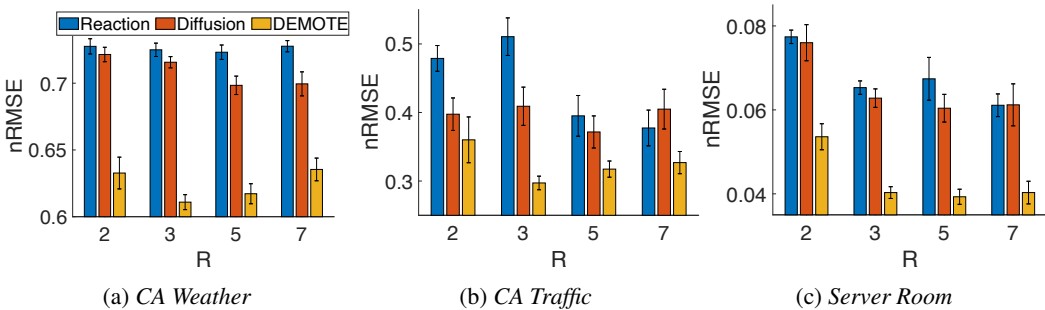

Figure 3: Predictive performance of the diffusion and reaction processes.

mode tensor (*traffic type*, *severity level*, *latitudes*, *longitude*). There are 7 traffic types, 6 severity levels, 20 latitudes and 20 longitudes. We collected 30K entry values (accident counts) at different time points. (3) *Server Room* (https://zenodo.org/record/3610078#.XlNpAigzaM8), temporal temperature records of Poznan Supercomputing and Networking Center. The temperatures were measured at 34 locations, under different air-condition modes ($24°$, $27°$, and $30°$) and power usage settings (50%, 75% and 100%). Hence, we extracted a three-mode tensor (*location*, *air-condition mode*, *power level*). In total, 10K observed entry values and their timestamps were collected.

**Competing Methods.** The following popular and/or state-of-the-art temporal decomposition approaches were compared. (1) CP-DTLD, discrete-time CP decomposition with linear dynamics, where a conditional prior is placed over successive time steps, $p(\mathbf{t}_{j+1}|\mathbf{t}_j) = \mathcal{N}(\mathbf{t}_{j+1}|\mathbf{A}\mathbf{t}_j + \mathbf{b}, v\mathbf{I})$; $\mathbf{A}$, $\mathbf{b}$ and $v$ were jointly estimated during the CP decomposition. Note that (Xiong et al., 2010) is an instance of this model where $\mathbf{A} = \mathbf{I}$ and $\mathbf{b} = \mathbf{0}$. (2) GP-DTLD and (3) NN-DTLD, similar to CP-DTLD, except using GP (Zhe et al., 2016b) and NN decomposition models (similar to (Liu et al., 2019)), respectively. (4) CP-DTND, (5) GP-DTND and (6) NN-DTND — CP, GP and NN decomposition with nonlinear dynamics, where the conditional prior is $p(\mathbf{t}_{j+1}|\mathbf{t}_j) = \mathcal{N}(\mathbf{t}_{j+1}|\sigma(\mathbf{A}\mathbf{t}_j)+\mathbf{b}, v\mathbf{I})$ where $\sigma(\cdot)$ is a nonlinear activation. The dynamics can therefore be viewed as an RNN transition. (7) CP-CT (Zhang et al., 2021), continuous-time CP factorization, which models the CP coefficients as a time-varying function, with polynomial splines. (8) GP-CT, continuous-time GP decomposition that extends (Xu et al., 2012; Zhe et al., 2016b) by plugging the time in the GP kernel so as to estimate the entry value as a function of the embeddings and time, $m_{\boldsymbol{\ell}} = g(\mathbf{u}_{\ell_1}^1, \ldots, \mathbf{u}_{\ell_K}^K, t)$. (9) NN-CT, continuous-time NN decomposition, where the input consists of both the embeddings and time $t$. (10) THIS-ODE (Li et al., 2022), a continuous-time decomposition, where a neural ODE is used to estimate the tensor entry values given the static embeddings and time. (11) NONFAT (Wang et al., 2022), a bi-level latent GP model that uses Fourier bases to estimate factor trajectories for dynamic tensor decomposition.

**Settings and Results.** All the approaches were implemented with PyTorch. The Square Exponential kernel was used for all the GP-related methods, including GP-{DTLD, DTND, CT}. We used the same variational sparse approximation (Hensman et al., 2013) to fulfill scalable posterior inference. Following (Zhe et al., 2016b), the number of inducing point was set to 100. For the NN decomposition methods, we employed a three-layer network with `tanh` activation, and for THIS-ODE, we used a one-layer network. The layer width was chosen from {10, 25, 50, 75, 100}. We used `tanh` as the activation function in the nonlinear dynamic baselines, including {CP, GP, NN}-DTND. For our method, we used the same NN architecture for both the reaction process and entry value prediction, which includes two hidden layers with 50 neurons per layer. For CP-CT, we employed 100 knots to fulfill the polynomial splines. For each discrete-time method, the number of time steps was chosen from {25, 50, 75, 100} via the cross-validation on the training set. We trained all the models with stochastic mini-batch optimization. We used the ADAM algorithm, and the mini-batch size was set to 100. We ran every method with 10K epochs to ensure convergence. The learning rate was automatically adjusted in $[10^{-4}, 10^{-1}]$ by the `ReduceLROnPlateau` scheduler. We varied the dimension of the embeddings $R$ from {2, 3, 5, 7}. For DEMOTE, $R$ is the number of embedding trajectories; we used computational graphs to obtain the gradient. We followed (Kang et al., 2012; Zhe et al., 2016b) to randomly draw $80\%$ observed entries and their time stamps for training, with the remaining for test. We computed the normalized root-mean-square error (nRMSE). We repeated the evaluation five times and computed the average nRMSE and standard deviation.

| CA Weather | $R = 2$ | $R = 3$ | $R = 5$ | $R = 7$ |
|---|---|---|---|---|
| CP-DTLD | $0.7440 \pm 0.0035$ | $0.7372 \pm 0.0040$ | $0.7290 \pm 0.0042$ | $0.7270 \pm 0.0044$ |
| GP-DTLD | $0.7417 \pm 0.0031$ | $0.7414 \pm 0.0036$ | $0.7444 \pm 0.0036$ | $0.7449 \pm 0.0039$ |
| NN-DTLD | $0.7228 \pm 0.0054$ | $0.7116 \pm 0.0033$ | $0.7070 \pm 0.0041$ | $0.7065 \pm 0.0038$ |
| CP-DTND | $0.7448 \pm 0.0031$ | $0.7360 \pm 0.0035$ | $0.7273 \pm 0.0037$ | $0.7280 \pm 0.0044$ |
| GP-DTND | $0.7399 \pm 0.0034$ | $0.7346 \pm 0.0032$ | $0.7448 \pm 0.0037$ | $0.7467 \pm 0.0031$ |
| NN-DTND | $0.7113 \pm 0.0045$ | $0.6979 \pm 0.0126$ | $0.6659 \pm 0.0122$ | $0.6543 \pm 0.0155$ |
| CP-CT | $1.0000 \pm 0.0096$ | $0.9959 \pm 0.0067$ | $1.0010 \pm 0.0017$ | $1.0060 \pm 0.0034$ |
| GP-CT | $0.7433 \pm 0.0038$ | $0.7354 \pm 0.0027$ | $0.7359 \pm 0.0034$ | $0.7377 \pm 0.0033$ |
| NN-CT | $0.8697 \pm 0.0014$ | $0.8679 \pm 0.0022$ | $0.8676 \pm 0.0018$ | $0.8695 \pm 0.0016$ |
| NONFAT | $0.7444 \pm 0.0042$ | $0.7460 \pm 0.0032$ | $0.7645 \pm 0.0061$ | $0.7553 \pm 0.0029$ |
| THIS-ODE | $0.7511 \pm 0.0052$ | $0.7539 \pm 0.0041$ | $0.7614 \pm 0.0024$ | $0.7620 \pm 0.0032$ |
| DEMOTE | $\mathbf{0.6327 \pm 0.0119}$ | $\mathbf{0.6109 \pm 0.0056}$ | $\mathbf{0.6172 \pm 0.0075}$ | $\mathbf{0.6354 \pm 0.0085}$ |
| CA Traffic | | | | |
| CP-DTLD | $0.6498 \pm 0.0257$ | $0.6424 \pm 0.0266$ | $0.6436 \pm 0.0268$ | $0.6405 \pm 0.0262$ |
| GP-DTLD | $0.6309 \pm 0.0167$ | $0.6290 \pm 0.0185$ | $0.6383 \pm 0.0204$ | $0.6496 \pm 0.0193$ |
| NN-DTLD | $0.6528 \pm 0.0230$ | $0.6545 \pm 0.0244$ | $0.6401 \pm 0.0282$ | $0.6136 \pm 0.0338$ |
| CP-DTND | $0.6497 \pm 0.0245$ | $0.6456 \pm 0.0265$ | $0.6431 \pm 0.0263$ | $0.6419 \pm 0.0259$ |
| GP-DTND | $0.6544 \pm 0.0213$ | $0.6559 \pm 0.0224$ | $0.6604 \pm 0.0243$ | $0.6674 \pm 0.0214$ |
| NN-DTND | $0.6578 \pm 0.0248$ | $0.6528 \pm 0.0256$ | $0.6519 \pm 0.0249$ | $0.6482 \pm 0.0261$ |
| CP-CT | $0.9858 \pm 0.0120$ | $0.9972 \pm 0.0056$ | $0.9816 \pm 0.0136$ | $0.9991 \pm 0.0120$ |
| GP-CT | $0.6610 \pm 0.0207$ | $0.6668 \pm 0.0191$ | $0.6756 \pm 0.0190$ | $0.6768 \pm 0.0196$ |
| NN-CT | $0.9804 \pm 0.0017$ | $0.9815 \pm 0.0015$ | $0.9791 \pm 0.0012$ | $0.9802 \pm 0.0017$ |
| NONFAT | $0.4461 \pm 0.0247$ | $0.4610 \pm 0.0231$ | $0.5031 \pm 0.0155$ | $0.6307 \pm 0.0847$ |
| THIS-ODE | $0.6603 \pm 0.0230$ | $0.6536 \pm 0.0212$ | $0.6838 \pm 0.0193$ | $0.6378 \pm 0.0142$ |
| DEMOTE | $\mathbf{0.3601 \pm 0.0334}$ | $\mathbf{0.2972 \pm 0.0099}$ | $\mathbf{0.3174 \pm 0.0118}$ | $\mathbf{0.3269 \pm 0.0162}$ |
| Server Room | | | | |
| CP-DTLD | $0.4211 \pm 0.0029$ | $0.4209 \pm 0.0031$ | $0.4208 \pm 0.0028$ | $0.4208 \pm 0.0028$ |
| GP-DTLD | $0.0914 \pm 0.0020$ | $0.0791 \pm 0.0010$ | $0.0739 \pm 0.0014$ | $0.0753 \pm 0.0013$ |
| NN-DTLD | $0.4213 \pm 0.0032$ | $0.4213 \pm 0.0032$ | $0.4212 \pm 0.0034$ | $0.4205 \pm 0.0030$ |
| CP-DTND | $0.2835 \pm 0.0160$ | $0.1751 \pm 0.0020$ | $0.1174 \pm 0.0011$ | $0.0829 \pm 0.0044$ |
| GP-DTND | $0.0925 \pm 0.0013$ | $0.0784 \pm 0.0011$ | $0.0739 \pm 0.0009$ | $0.0774 \pm 0.0009$ |
| NN-DTND | $0.4213 \pm 0.0032$ | $0.4212 \pm 0.0030$ | $0.4211 \pm 0.0032$ | $0.4205 \pm 0.0030$ |
| CP-CT | $0.9919 \pm 0.0096$ | $0.9951 \pm 0.0050$ | $0.9862 \pm 0.0109$ | $1.0121 \pm 0.0070$ |
| GP-CT | $0.1385 \pm 0.0020$ | $0.1223 \pm 0.0016$ | $0.1275 \pm 0.0014$ | $0.1365 \pm 0.0014$ |
| NN-CT | $0.1193 \pm 0.0030$ | $0.1140 \pm 0.0015$ | $0.1113 \pm 0.0027$ | $0.1149 \pm 0.0028$ |
| NONFAT | $0.1468 \pm 0.0026$ | $0.1407 \pm 0.0023$ | $0.1396 \pm 0.0022$ | $0.1409 \pm 0.0030$ |
| THIS-ODE | $0.1412 \pm 0.0024$ | $0.1312 \pm 0.0013$ | $0.1304 \pm 0.0016$ | $0.1350 \pm 0.0019$ |
| DEMOTE | $\mathbf{0.0536 \pm 0.0031}$ | $\mathbf{0.0403 \pm 0.0014}$ | $\mathbf{0.0393 \pm 0.0018}$ | $\mathbf{0.0403 \pm 0.0027}$ |

Table 1: Normalized Root Mean-Square Error (nRMSE). The results were averaged from five runs.

As shown in Table 1, DEMOTE consistently achieves the best prediction accuracy and in many cases outperforms the competing methods by a large margin. Although learning an embedding trajectory is much more challenging than learning a fixed-value embedding, the experimental results have demonstrated the advantage of our method in predictive performance. To investigate the effect of the two processes in our model, we also examined our method with the diffusion process only and with the reaction process only on all the datasets. Their performance, as compared with DEMOTE, is shown in Fig. 3. We can see that each individual component can lead to good prediction accuracy. However, each component is worse than DEMOTE that synergizes the two components together. Therefore, the results show that each process is effective, and more important, the two processes can bolster each other to further improve the performance when they are combined.

## 6.3 Learning Result Investigation

Next, we looked into the learned embedding trajectories and checked if they exhibit patterns. To do so, we set $R = 3$ and ran DEMOTE on *Server Room* dataset. In Fig. 4, we show the learned embedding trajectories for the first location (a-c), the first air condition mode (d-f) and the first power usage level (g-i). As we can see, even for the same object, *e.g.*, a particular location, the corresponding embedding trajectories vary quite differently, implying the evolution of different underlying properties, such as the workload, memory usage, and network latency. We further found there are underlying structures within the embeddings during their evolution. We listed the results in Appendix (Sec. A).

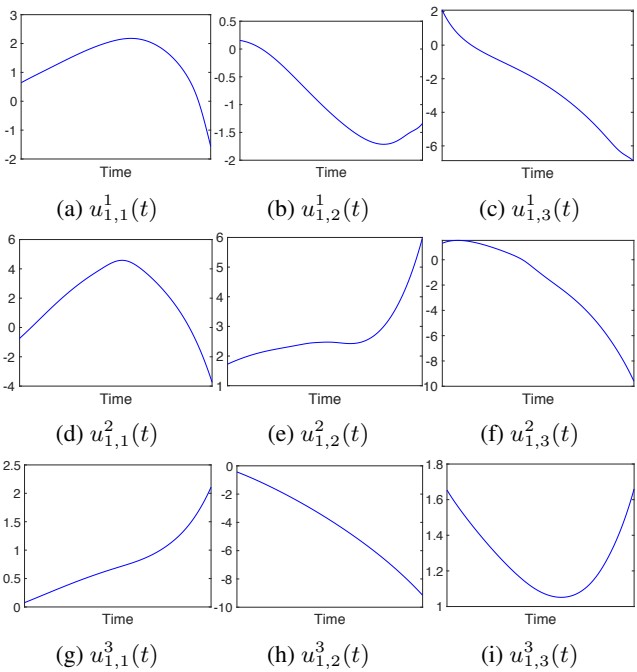

(a) $u_{1,1}^1(t)$      (b) $u_{1,2}^1(t)$      (c) $u_{1,3}^1(t)$

(d) $u_{1,1}^2(t)$      (e) $u_{1,2}^2(t)$      (f) $u_{1,3}^2(t)$

(g) $u_{1,1}^3(t)$      (h) $u_{1,2}^3(t)$      (i) $u_{1,3}^3(t)$

Figure 4: The learned embedding trajectories for location 1 (a-c), air conditional mode 1 (d-f), and power usage level 1 (g-i) in *Server Room* dataset.

We showcase the temporal predictions for two tensor entries. As we can see from Fig. 5, given only a few training points (blue), our method can predict the test points (green) much more accurately, as compared with GPCT, and the predictive uncertainty (reflected by the noise variance $\sigma^2$) is much smaller. This might be due to that via the diffusion-reaction process, and our method can more effectively extract the temporal knowledge from sparse data. For example, DEMOTE successfully captured the periodic nature in the first entry (Fig. 5b) while GPCT treated the fluctuation as noises and ended up with much inaccurate predictions and larger predictive variances.

**Computational Efficiency.** We compared the per-epoch/iteration running time DEMOTE with the other methods. We tested all the methods in a workstation with one NVIDIA GeForce RTX 3090 Graphics Card, 10th Generation Intel Core i9-10850K Processor, 32 GB RAM, and 1 TB SSD. The results are shown in Table 2 in Appendix. We can see that the running speed of DEMOTE is comparable to NONFAT and other NN decomposition methods. We also compared with running DE-MOTE with naive sampling (DEMOTE-NS). The stratified sampling led to 4x to 22x speed-up.

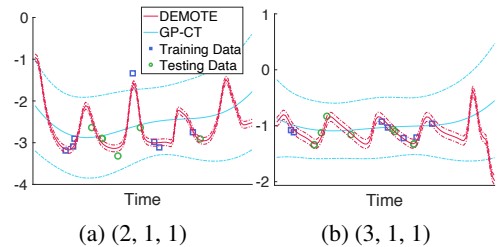

(a) (2, 1, 1)      (b) (3, 1, 1)

Figure 5: Entry value prediction on *Server Room*.

## 7 Conclusion

We have presented DEMOTE, a neural diffusion-reaction process model to learn dynamic embeddings for dynamic tensor decomposition. The predictive performance is encouraging, and the learned embedding trajectories exhibit interesting patterns. Currently, our method is limited to a small number of entities since it has to integrate the entire multi-partite graph to construct the diffusion process. In the future work, we plan to develop graph cut algorithms to partition the graph into a set of small sub-graphs so that we can construct multiple diffusion processes in parallel so as to scale up our model to big graphs and to large tensors.

## Acknowledgments

This work has been supported by NSF CAREER Award IIS-2046295.

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

# Appendix

## A   Investigation of Embedding Dynamics

We investigated if there are underlying structures within the embeddings during their evolution. To this end, we looked into the embeddings of the 34 locations on *Server Room* dataset at five time points ($t = 1, 20, 50, 80, 100$). At each time point, we ran the k-means algorithm over the embeddings to extract the clustering structures. We used the elbow method (Ketchen and Shook, 1996) to select the number of clusters. We can see that at earlier time ($t \leq 50$), the clusters are more compact, while at the later stages, the clusters become more scattered. This reflects how the structure of those entities (*i.e.*, locations) evolves along with time. It is interesting to see that some locations are in the same cluster all the time, like location {5,7} and location {16, 32}. It implies that their underlying properties might have quite similar (or correlated) evolution patterns. Some locations are grouped in the cluster at the beginning, *e.g.*, location {32, 34} (at $t = 1$), but later moves to different clusters ($t > 1$). It implies their evolution patterns can vary significantly, leading to the change of the cluster memberships.

|  | CA Weather | CA Traffic | Server Room |
|---|---|---|---|
| CP-DTLD | 0.037 | 0.086 | 0.023 |
| GP-DTLD | 0.246 | 0.247 | 0.248 |
| NN-DTLD | 2.400 | 4.730 | 1.080 |
| CP-DTND | 0.038 | 0.087 | 0.025 |
| GP-DTND | 0.119 | 0.242 | 0.080 |
| NN-DTND | 2.360 | 4.701 | 1.060 |
| CP-CT | 0.025 | 0.052 | 0.018 |
| GP-CT | 0.068 | 0.216 | 0.105 |
| NN-CT | 2.310 | 3.885 | 1.030 |
| NONFAT | 0.952 | 1.925 | 0.571 |
| THIS-ODE | 58.710 | 136.100 | 7.190 |
| DEMOTE | 1.390 | 1.895 | 0.309 |
| DEMOTE-NS | 6.12 | 10.42 | 7.06 |

Table 2: Per-epoch/iteration running time (in seconds). DEMOTE-NS means running DEMOTE with naive sampling of min-batches rather than the stratified sampling.

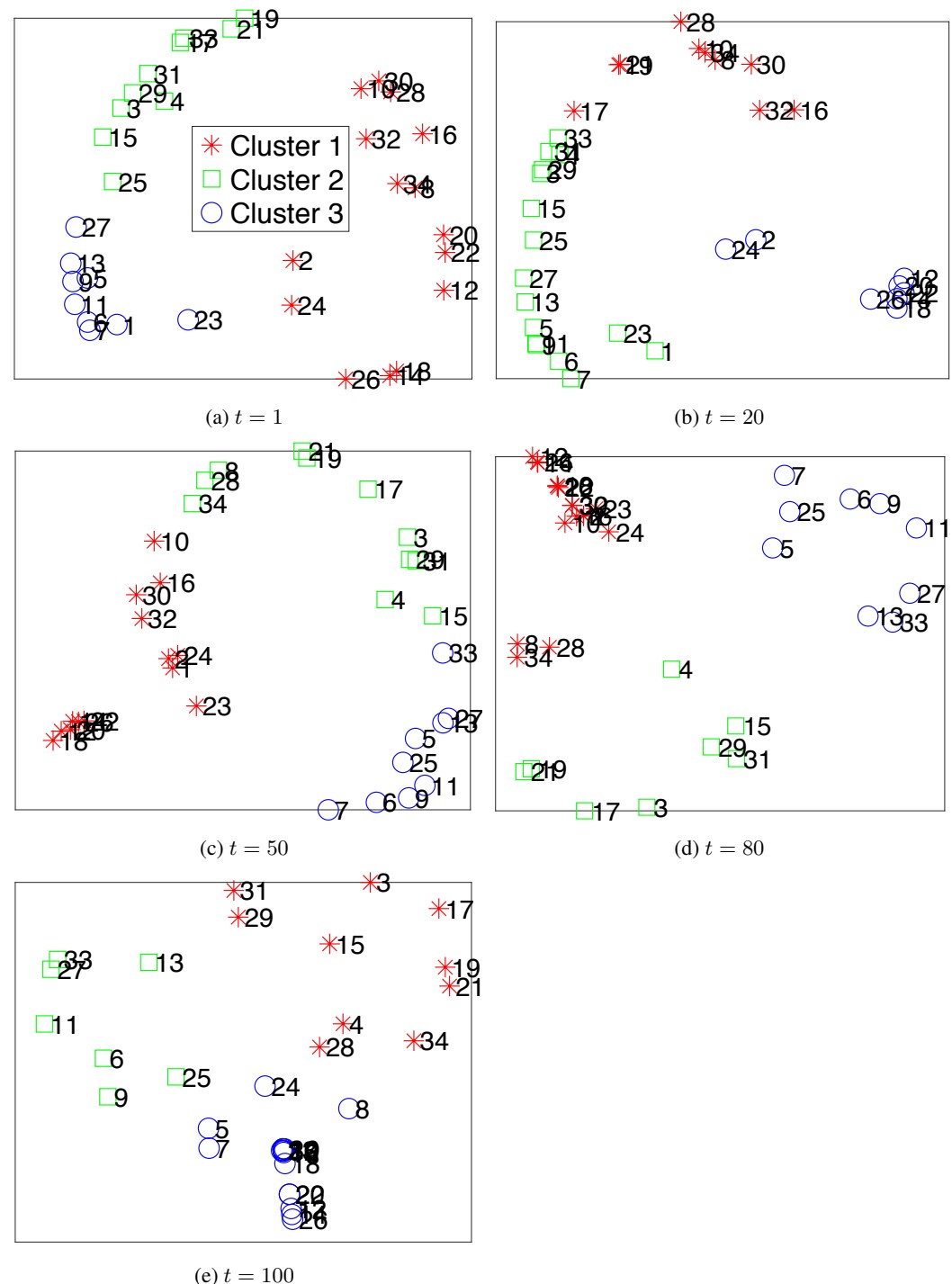

Figure 6: Evolution of the clustering structure within the 34 locations on *Server Room* dataset.

