# OpenReview forum: "Dynamic Tensor Decomposition via Neural Diffusion-Reaction Processes"
_NeurIPS.cc/2023/Conference — NeurIPS 2023 spotlight_

### Official Review · Reviewer_wffj · 2023-07-01

**Soundness:** 2 fair
**Presentation:** 2 fair
**Contribution:** 3 good
**Rating:** 6
**Confidence:** 3

**Summary:**

The paper presents the framework to generate dynamic embeddings for multi-entity (e.g., traffic type, severity, latitude, and longitude) interactions with timestamps. The temporal high-order interactions are modeled via neural diffusion-reaction processes, and to tack the temporal drift of underlying node, authors propose the dynamic embeddings that follows a diffusion-based ODE model. The diffusion process uses a multi-partite graph to coevolve the trajectories, and the reaction process models the individual evolution of each object. The combination of these two processes can capture both the commonalities and personalities in learning the embedding trajectories for different objects. Authors demonstrate that the proposed model can capture the temporal dynamic well through both synthetic and real-world datasets with comparison to a large number of existing methods.

**Strengths:**

1. The proposed ODE-based model looks like a reasonable way of representing the dynamics of embeddings.

2. The dynamic embeddings for entities are generated in continuous, rather than discrete time, unlike most existing approaches.

3. Numerical experiments demonstrate that the proposed method outperforms baselines on 3 real-world datasets.

**Weaknesses:**

1. It seems that the text can be further improved to present better. Some phrases are repeated several times unnecessarily, and some points, on the contrary, are covered too briefly. I would recommend the authors to carry out additional proofreading of the text.

2. The source code is not provided, and this complicates the reproducibility of the results.

**Questions:**

1. [Line 6] It seems that "I" in "EMbedIngs" should be lowercase.

2. [Lines 92-100] It seems that there are not enough conclusions to this paragraph. What are the advantages and disadvantages of the indicated "continuous CP decomposition" approach?

3. [Line 106] "typical" -> "typically".

4. [Line 270] "We compared <our approach>..."

**Limitations:**

The proposed method is limited to a small number of entities, however, further development of the approach using the graph cut algorithms could potentially improve its scalability.

---

> ### Author Rebuttal · Authors · 2023-08-07
>
> Thanks for your valuable comments and suggestions. Here are our responses. C: comments; R: responses
>
> > C1: "It seems that the text can be further improved to present better. Some phrases are repeated several times unnecessarily, and some points, on the contrary, are covered too briefly. I would recommend the authors to carry out additional proofreading of the text."
>
> R1: Great suggestion and we do agree. We will polish our paper to further improve the readability and clarify.
>
> > C2: "The source code is not provided"
>
> R2: We appreciate the reviewer's concern. We have provided the code via an anonymous link (see the global response R1).
>
> > C3: "[Lines 92-100] It seems that there are not enough conclusions to this paragraph. What are the advantages and disadvantages of the indicated "continuous CP decomposition" approach?"
>
> R3: Great question. The continuous-time CP decomposition integrates the continuous time information into the CP framework, and hence will not lose information like other methods that simply bin the timestamps into coarse steps (see the references in Line 94-95). However, the disadvantage of this approach is that it still estimates a static embedding and cannot capture the evolution of the representations (see the motivation of our work presented in Line 102-109). In addition, CP is a simple multilinear decomposition model, and is not flexible enough to capture nonlinear interactions of the entities. Hence, our work uses a neural network decomposition model instead (see Eq2 and Line 160-166). We will supplement this discussion into the paper.

---

> > ### Comment · Reviewer_wffj · 2023-08-13
> >
> > Thank you for your comments. Provided that you proofread the text of the paper and place in the text a link to the repository with the code when publishing, my main remarks appear to be removed. I still have concerns about the limitations of your method that I mentioned in the review, but your work seems interesting to the scientific community and well-researched, so I'm raising my rating from 5 to 6. Good luck!

---

> > > ### Author Response · Authors · 2023-08-13
> > >
> > > We appreciate your feedback and thanks for raising the rating!

---

### Official Review · Reviewer_3waT · 2023-07-06

**Soundness:** 3 good
**Presentation:** 3 good
**Contribution:** 2 fair
**Rating:** 6
**Confidence:** 4

**Summary:**

In this work, the authors introduce DEMOTE which can both consider both structure information and temporal information via dynamic tensor decomposition. Structure information is captured by multi-partite graph diffusion process which in substance is a special ODE, whereas temporal information is analyzed using another ODE. Experimental results demonstrate DEMOTE in general outperforms one baseline for synthetic data and many baselines for real-world data.

**Strengths:**

1. It is interesting to model the spatial interactions for values at the same time point using multi-partite graphs through graph diffusion, for the small-scale datasets.

2. Addressing the sparsity in tensorial data is another strength of this work. It has been an important problem in tensorial data and appeal to the tensor learning community.

**Weaknesses:**

1. There have been some existing work on dynamic tensor decomposition using neural ODEs and relational information in the recent years, whereas not discussed and compared in this work, for example, TANGO [1].

2. The presentation should be improved. The notations should be defined in a clear manner, for example, the adjacency tensor W needs more clarification on its order and how it is computed, from its indices.

Reference

[1] Han, Z., Ding, Z., Ma, Y., Gu, Y., & Tresp, V. (2021, November). Learning neural ordinary equations for forecasting future links on temporal knowledge graphs. In *Proceedings of the 2021 conference on empirical methods in natural language processing* (pp. 8352-8364).

**Questions:**

1. For the statement in Page 3 Section 3 Lines 113-115, why can learning dynamics produce more reliable and long-term trajectory predictions? The time derivative can provide the general dynamics of the whole time series and mainly focus on the global sequential information. Please check this paper "Neural Rough Differential Equations for Long Time Series" [1].

2. Could the authors provide any explanation and examples on why K should be involved and represent structure information using multi-partite graphs is necessary? It seems that including the multi-partite graphs is very complicated, and in what circumstances the multi-partite graphs should be applied to describe what types of intercorrelation among multi-dimensional data, without incurring overfitting?

3. In Figure 2(c), why do the red lines decline? In the long-term, the performance of DEMOTE is worse than the baseline model NONFAT. Could the authors articulate it?

4. For the results of DEMOTE in Figure 5(a) and (b), why are the entry value predictions with high and sharp spikes across time? It seems that DEMOTE can capture the periodic fluctuations, but DEMOTE by nature is based on two ODEs. In specific, ODE and neural ODE assume that the modeled trajectory should be smooth, or at least Lipschitz continuous [2,3]. Could the authors also provide their code?

References

[1] Morrill, J., Salvi, C., Kidger, P., & Foster, J. (2021, July). Neural rough differential equations for long time series. In International Conference on Machine Learning (pp. 7829-7838). PMLR.

[2] Chen, R. T., Rubanova, Y., Bettencourt, J., & Duvenaud, D. K. (2018). Neural ordinary differential equations. Advances in neural information processing systems, 31.

[3] Süli, E., & Mayers, D. F. (2003). An introduction to numerical analysis. Cambridge university press.

**Limitations:**

The limitations are sufficiently addressed by the authors.

---

> ### Author Rebuttal · Authors · 2023-08-07
>
> Thanks for your insightful comments and the great reference TANGO. We will add discussion and comparison with this nice, related work.
>
> >C1: why can learning dynamics produce more reliable and long-term trajectory predictions?
>
> R1: Great question. To better illustrate it, we can consider an alternative popular model, Gaussian process (GP), to directly learn the trajectories. GP is known to work well when making prediction near the training inputs. However, at places far from the training inputs, the prediction accuracy degrades severely, because the interpolation on the training points becomes ineffective (the similarity to the training inputs drops quickly). Hence, GP cannot provide reliable/long-term trajectory prediction. However, if we use ODE modeling, we learn the dynamics rather than try to interpolate the prediction. As the reviewer pointed out,  "the time derivative can provide the general dynamics of the whole time series". Hence, as long as we can well capture the time derivative --- this is what ODE modeling is good at --- we can use it to infer long-term trajectories reliably, no matter if the places are close to or far away from the training inputs.
>
> In fact, our intuition has been verified by the previous work (Rubanova et al., 2019)  that builds latent ODE models for time series (cited at Line 42). In Fig. 4a and 5b of their paper, they show their ODE models can use a small amount of training observations to well extrapolate to a long time horizon; also see Table 3 their much better quantitative results in extrapolation (Enc-Dec).
>
> Third, thanks for the great reference (Morrill et al., 2021). But this work is about **Neural Control Differential Equation models, i.e., Neural CDE rather than Neural ODE**. Due to a different model design (Neural CDE is a continuous version of RNN while Neural ODE is a continuous version of ResNet), Neural CDEs have an additional, significant learning challenge: one has to densely approximate the time derivative of the observed sequence, which is also integrated into the dynamics. According to their paper, "Neural CDEs, as with RNNs, begin to break down for long time series. Loss/accuracy worsens, and training time becomes prohibitive due to the sheer number of forward operations within each training epoch". However, Neural ODEs do not have to face such challenge, and the bottleneck might not apply to them.
>
> We will add more detailed discussions about the model choice in the paper.
>
> >C2: "any explanation and examples on why K should be involved and represent structure information using multi-partite graphs is necessary? ..."
>
> R2: Great questions. First, tensor data are about multiway interactions, e.g., *(customer, seller, product)*, and the entry value is the interaction result, e.g., payment. The presence of these interactions indicate intercorrelations between the involved entities (in each mode). We are motivated to leverage these correlations to enhance learning embedding trajectories in our ODE framework. One elegant way is to design a diffusion process, which *needs to be on a graph*. We therefore convert the observed multiway interactions into a K-partite graph. We believe this graph is reasonable to encode various correlations in the data. For example, if any two entities appear in the same observed interaction (tensor entry), we add an edge between them to indicate their inter-correlation. Note that we do not add edges between the entities in the same mode, because a tensor entry always represents the interaction between different modes. That's why our graph is a multi-partite graph. See Line 122-135 for more detailed explanation/rationale for our graph construction. With the graph, we are then able to design our diffusion-reaction process (Eq1).
>
> Second, we can choose to remove the graph and hence remove the diffusion process in learning. However, this will treat the embedding trajectory of each entity independently and bring the risk of overfitting. The K-partite graph and the associated diffusion process not only bring additional structural knowledge to facilitate learning, but also induces a proper regularization (homophily) to alleviate overfitting. In fact, our experiments (see Fig. 3) have shown that only learning with the reaction process (no diffusion) performs much worse. Please see R1 for Reviewer 9boz for more detailed discussion.
>
> Third, we agree that the K-partite graph might not be the only choice. There might be other possible graph representations of the observed data. This is an interesting open equation. We will investigate it in the future study.
>
> >C3: "In Fig. 2(c), why do the red lines decline? In the long-term, the performance of DEMOTE is worse than the baseline model NONFAT. Could the authors articulate it?"
>
> R3: Actually, Fig. 2c is **NOT** about the prediction accuracy. The decline of the red line does **NOT** mean worse long-term performance. The whole Fig. 2 shows the embedding trajectories learned by different methods (see the caption). Fig. 2a & 2d show the ground-truth, Fig. 2b & 2e show the trajectories learned by NONFAT while Fig. 2c & 2f the learned trajectories by DEMOTE. Comparing Fig. 2c and 2a, we can see that the learned trajectories by DEMOTE is much closer to the ground-truth than those learned by NONFAT (see Fig. 2b). Hence, the performance of DEMOTE actually is better than NONFAT.
>
> >C4: "For the results in Figure 5(a) and (b), why are the entry value predictions with high and sharp spikes across time?..."
>
> R4: Great question. Our method does NOT model the entry value as an ODE. We model the embedding trajectories in each mode as an ODE; see Eq1. Given the trajectory values, the entry value at a specific time point is then predicted by a neural network (rather than an ODE); see Eq 2. Therefore, it might not be surprising to see fluctuations at the final outcomes, i.e., entry value prediction, since our model only learns smooth trajectories in the latent embedding space (see Fig. 4).

---

> ### Author Response · Authors · 2023-08-19
>
> Dear Reviewer  3waT: Could we kindly know if the responses have addressed your concerns and if further explanations or clarifications are needed? Your time and efforts in evaluating our work are appreciated greatly.

---

> > ### Comment · Reviewer_3waT · 2023-08-21
> >
> > Thank you for your response. It answers all my questions. I would like to raise my score by 1.

---

> > > ### Author Response · Authors · 2023-08-21
> > >
> > > Thanks a lot. We appreciate your feedback!

---

### Official Review · Reviewer_gF6D · 2023-07-06

**Soundness:** 3 good
**Presentation:** 2 fair
**Contribution:** 2 fair
**Rating:** 6
**Confidence:** 3

**Summary:**

In this paper, to capture higher-order interaction among different modes of tensor data and their evolution process over time, the authors proposed an ODE-based method called DEMOTE to capture the dynamics of latent embeddings.   In detail, the authors propose to use a diffusion-reaction process to model the evolution of these embeddings.  What's more, structure knowledge within tensor data is considered in this diffusion-reaction process, which enhances the model's representation ability.  Experiments show the effectiveness of the proposed method.

**Strengths:**

1. The experimental results show an advantage compared with related work.

2. The way of using the structure knowledge is novel and effective in modeling tensor data. It is particularly interesting to embed this structured knowledge with the framework of ODE.

**Weaknesses:**

1. The main concern lies in its novelty since there are some efforts studying the interaction results of higher order continuous time data, as clarified in the paper (that is [1]). Actually, in [1], they use ODE to model the entry values' trajectories and depend on $m_{i}$. In this work, the authors use ODE to model $m_{i}$ directly. Although these two papers correspond to different ways of handling problems and the authors have stated the advantage of the proposed method, they do share many similar intuitions towards the same task, which hinders their novelty.

Typo: Page 3, last paragraph: $(v_{j}^{k}, v_{j'}^{k'})\in \mathbb{E}$ $\rightarrow$  $(v_{j}^{k}, v_{j'}^{k'})\in E$.

[1] Li S, Kirby R, Zhe S. Decomposing Temporal High-Order Interactions via Latent ODEs[C]//International Conference on Machine Learning. PMLR, 2022: 12797-12812.

**Questions:**

1. I think the contribution should be further clarified.

**Limitations:**

Yes, the authors have adequately addressed the limitations and potential negative societal impact of their work.

---

> ### Author Rebuttal · Authors · 2023-08-07
>
> Thanks for your valuable comments. Here are our responses. C: comments; R: responses
>
> > C1: "The main concern lies in its novelty since there are some efforts studying the interaction results of higher order continuous time data, as clarified in the paper (that is [1]) ... Although these two papers correspond to different ways of handling problems and the authors have stated the advantage of the proposed method, they do share many similar intuitions towards the same task, which hinders their novelty" "I think the contribution should be further clarified."
>
> R1: We appreciate the reviewer's concern. We do agree that both our work and [1] use ODE as  the modeling framework to handle temporal interaction data. While we have clarified the difference in our paper (Line 226-228), here we would like to give more detailed and comprehensive clarification. In fact, the task, model design, algorithm and evaluation of [1] are all very different from our work.
>
> First, the task/goal of [1] is **NOT** to estimate dynamic embeddings; it still estimates **static** embeddings (time-invariant), like most tensor decomposition methods do. The latent ODE model of [1] is only designed to predict the temporal interaction result. By contrast, the task of our work is to learn the embedding trajectories to capture the temporal evolution of the representations; we consider the embedding as a function of time. We believe this is scarce in existing works and our work fills an important gap in tensor decomposition.
>
> Correspondingly, a large portion of our experiment is to examine if the embedding trajectory recovered by our method is accurate (see Section 6.1 and Fig. 2 for Simulation Study) and meaningful/interpretable (see Section 6.3 Fig. 4, and Fig. 1 of Appendix). Such evaluations have never shown up in [1].
>
> Third, [1] has never leveraged the structural knowledge underlying the observed interactions; [1] constructs a neural ODE model in a straightforward manner. The key hypothesis of our work is that such knowledge is important for trajectory learning. We proposed a novel k-partite graph to represent the structure, based on which we proposed a neural diffusion-reaction process to integrate the structural knowledge into the embedding trajectory learning and interaction result prediction.
>
> Fourth, [1] uses forward sensitivity for model training. That is, [1] augments the ODE model with the sensitivity (i.e., gradient), and performs a forward solve to jointly compute the ODE states and state gradient. This is feasible, because [1] models each entry value as an ODE, and the state is just a scalar (one dimensional). However, the learning algorithm of [1] does not apply to our work, because our ODE state includes all the embedding trajectories and is high-dimensional. For efficient learning, we devised a stratified mini-batch sampling method, and combined with the adjoint state method (backward) or computational graph to compute the gradient for training.
>
> Thanks for bringing up the question. We will add more discussions about [1] and clarification about the difference of our work.

---

> ### Author Response · Authors · 2023-08-19
>
> Dear Reviewer gF6D:
> Could we kindly know if the responses have addressed your concerns and if further explanations or clarifications are needed? Your time and efforts in evaluating our work are appreciated greatly.

---

> > ### Comment · Reviewer_gF6D · 2023-08-21
> >
> > The authors' rebuttal appropriately addresses my main questions. I believe this work would contribute to the community in the future. In light of this, I would raise my score.

---

> > > ### Author Response · Authors · 2023-08-21
> > >
> > > Thank you. We appreciate your feedback!

---

### Official Review · Reviewer_9boz · 2023-07-06

**Soundness:** 3 good
**Presentation:** 3 good
**Contribution:** 2 fair
**Rating:** 6
**Confidence:** 3

**Summary:**

This paper proposes a dynamic tensor decomposition (TD) model, in which the latent factors are functions of time stamps described by neural ODEs. Moreover, they encode an adjacent matrix into the diffusion process to better model the correlations among different tensor modes. Finally, they use an additional neural network to map latent factors to tensor entries. Using differentiable ODE solvers, all the parameters can be learned by optimizing the joint likelihood with SGD. For empirical evaluations, the authors first conduct a simulation study to show that the proposed model can learn latent factor trajectories. Then, they demonstrate the completion performance on several dynamic tensor datasets.

**Strengths:**

1. The authors propose to use a diffusion process to learn tensor factor trajectories, which is new in the field.

2. The experiments show that the proposed model can learn factor and entry trajectories. Besides, it achieves a significant improvement on the completion performance.

**Weaknesses:**

1. In the paper, the authors construct a complex TD with ODEs and NN to parameterize the latent relation. Besides, the model parameters are learnt from MAP estimates (Eq. 3). I am wondering if such a model suffers from overfitting for sparse tensor data?

2. In Equation 1, the authors define the ODE of the factor trajectories. Since $\mathcal{F}(\mathcal{U}, t)$ is already parameterized by flexible neural networks, I am wondering if it is necessary to add the multi-partite graph term, i.e., $(\mathcal{W} - \mathcal{A})\mathcal{U(t)}$? Can we construct a specific network structure that is equivalent to Eq. 1?

**Questions:**

1. In the equation between Line 178 and 179, does $\mathcal{L}$ equal to $\log (Prior)$ **plus** the likelihood?

---

> ### Author Rebuttal · Authors · 2023-08-06
>
> We thank the reviewer for their valuable comments. Here are our responses. C: comments; R: response.
>
> >C1: "the authors construct a complex TD with ODEs and NN to parameterize the latent relation. Besides, the model parameters are learnt from MAP estimates (Eq. 3). I am wondering if such a model suffers from overfitting for sparse tensor data?"
>
> R1: Great question. We do agree solely using the NN components can bring the risk of overfitting. However, our work also constructs a multi-partitie graph, and integrates the graph into the embedding trajectory learning, namely, the diffusion process. As one major contribution of our work, the diffusion process not only incorporates the structural knowledge of the data, it also uses **the structural knowledge to regularize the learning of the embedding trajectories**. Specifically, it encourages homophily of the trajectories for neighboring nodes (as they are correlated) --- the change rate of $\mathbf{u}^k_j(t)$ is modeled to correspond to the difference with the neighborhood's trajectories; see Line 140-144 and the equation in between. Therefore, we believe the diffusion process can effectively inhibit overfitting for sparse data. Our experimental results have shown that our method performs much better than just learning each trajectory independently (with the same NN components, i.e., no diffusion, just reaction process); see Fig. 3.
>
> >C2: "In Equation 1, the authors define the ODE of the factor trajectories. Since $\mathcal{F}(\mathcal{U},t)$ is already parameterized by flexible neural networks, I am wondering if it is necessary to add the multi-partite graph term, i.e., $(\mathcal{W}-\mathcal{A})\mathcal{U}(t)$ Can we construct a specific network structure that is equivalent to Eq. 1?"
>
> R2: Thanks for the great question. Please see R1 for our explanation about why the multi-partite graph term is important --- we believe it not only leverages the structural knowledge to facilitate the trajectory learning, but also induces a proper regularization to avoid overfitting.
>
> Regarding ``a specific'' equivalent ''network structure'', yes, we can view the RHS of Eq1 as modeled by a Residual Network. That is, the output of this network is a linear transformation of $\mathcal{U}$ (i.e., the multi-partite graph term) **plus** an NN transform, namely $\mathcal{F}(\mathcal{U},t)$.
>
>
> >C3: "In the equation between Line 178 and 179, does $\mathcal{L}$ equal to $log(Prior)$ plus the likelihood?"
>
> R3: Yes, exactly. We appreciate the reviewer catching this typo. We will fix it in the paper.

---

> ### Author Response · Authors · 2023-08-19
>
> Dear Reviewer 9boz:
> Could we kindly know if the responses have addressed your concerns and if further explanations or clarifications are needed? Your time and efforts in evaluating our work are appreciated greatly.

---

> > ### Comment · Reviewer_9boz · 2023-08-20
> >
> > Thanks for the authors’ responses.
> > This work proposes a new trajectory learning model for tensors, which could be interesting to the field. I would like to raise my rating to 6.

---

> > > ### Author Response · Authors · 2023-08-20
> > >
> > > Thanks for your feedback!

---

### Author Rebuttal · Authors · 2023-08-07

We thank all the reviewers for their careful, insightful and constructive comments! Here we post a few global responses. C: comments; R: responses

>C1: provide the code

R1: Yes, we are happy to provide the code. We have created an anonymous repository that includes the implementation of our model and algorithm and scripts to run the experiments. According to the rule of author response, `"If you were asked by the reviewers to provide code, please send an anonymized link to the AC in a separate comment"`, we have sent the anonymized link to the AC.

>C2: improve the presentation

R2: We appreciate the reviewer's suggestion. We will improve the notations and presentation to further enhance the readability and clarity of our paper.

---

### Author Response · Authors · 2023-08-19

We appreciate all reviewers' time and efforts in evaluating our work! In view of the limited available time, we would kindly like to ask the reviewers to please engage in a discussion with us (if not) given the submitted rebuttals so we can respond to the possible new questions in time.

---

### Decision · Program_Chairs · 2023-09-21

**Decision:**

Accept (spotlight)

**Comment:**

This paper proposes a dynamic tensor decomposition (TD) model, in which the latent factors are functions of time stamps described by neural ODEs. It demonstrates that the proposed model can capture the temporal dynamic well through both synthetic and real-world datasets with comparison to a large number of existing methods. This work proposes a new trajectory learning model for tensors, which could be interesting to the field. The authors' rebuttal have successfully addressed some questions and minor issues.